# Dendrological Secrets of the Pazaislis Monastery in Central Lithuania: DNA Markers and Morphology Reveal *Tilia* × *europaea* L. Hybrids of an Impressive Age

**DOI:** 10.3390/plants12203567

**Published:** 2023-10-13

**Authors:** Girmantė Jurkšienė, Darius Danusevičius, Rūta Kembrytė-Ilčiukienė, Virgilijus Baliuckas

**Affiliations:** 1Lithuanian Research Centre for Agriculture and Forestry, Liepu Str. 1 Girionys, LT-53101 Kaunas, Lithuania; virgilijus.baliuckas@lammc.lt; 2Faculty of Forest Sciences and Ecology, Agriculture Academy, Vytautas Magnus University, K. Donelaičo g. 58, LT-44248 Kaunas, Lithuania; darius.danusevicius@vdu.lt (D.D.); ruta.kembryte@vdu.lt (R.K.-I.)

**Keywords:** lime, linden, hybridization, parks, DNA markers, dendrology, leaf morphology

## Abstract

We benefited from the availability of a species-specific DNA marker to describe the morphometry of *T. cordata* × *platyphyllos* hybrids of an impressive age (ca. 150 years) grown in the Pazaislis baroque monastery yard in Central Lithuania. In an earlier study on a country-wide set of 543 *T. cordata* individuals from natural forest populations in Lithuania, we detected a nuclear microsatellite locus Tc8 well-differentiating between *T. cordata* and *T. platyphyllos*. The Tc8 locus contained a 140 bp allele in *T. cordata* (541 sampled individuals) and alleles above 160 bp in the two trees with a *T. platyphyllos*-like morphology (sampled in a national park). To verify the Tc8 locus as species specific, we sampled a further four *T. platyphyllos*-like individuals, which all contained the Tc8 locus alleles above 160 bp. We subsequently genotyped the six old-growth individuals from the Pazaislis monastery with mixed *T. cordata* × *platyphyllos* morphology. Results revealed that all six old-growth *Tilia* individuals from the Pazaislis monastery were heterozygous for the Tc8 locus with alleles of 140 bp (indicative of *T. cordata*) and 162 bp (indicative of *T. platyphyllos*). This finding confirms the morphological observations that these individuals are hybrids between *T. cordata* and *T. platyphyllos*. Additionally, the genotyping of a set of 14 nuclear microsatellite loci revealed that all six trees from the Pazaislis monastery are clones, possessing identical microsatellite genotypes. After the molecular identification, we morphotyped leaves, bracts, twigs, and nuts of the 6 old-growth *T. cordata* × *platyphyllos* hybrids from the Pazaislis monastery, 16 *T. cordata* old-growth trees, 4 *T.* × *europaea* var. *europaea* ‘Pallida’ trees growing near the Pazaislis monastery, and 4 mature *T. platyphyllos* trees from a nearby Girionys park. The morphotyping showed that *T. cordata* × *platyphyllos* hybrids may be the easiest to distinguish from *T. cordata* by raised and horizontally tertiary veins of leaves.

## 1. Introduction

For many centuries, linden trees have been planted for forestry and ornamental purposes in northern Europe [1]. Among commonly used species were *Tilia cordata* Mill., *T. platyphyllos* Scop., and *T*. × *europaea* L. (the latter was the *T. cordata* × *platyphyllos* hybrid). Namely, the hybrids of various reciprocities exhibit a variety of morphological features, in this way inhibiting the correct dendrological identification of linden individuals. Finding robust morphological markers for the correct taxonomic identification of *T*. × *europaea* L. constitutes the main scientific problem addressed in our study.

Linden is a valuable forest and ornamental tree whose history is intertwined with the landscape, folklore, avenues of cities and parks, and beekeeping in many European countries [1,2]. Small-leaved linden (*Tilia cordata* Mill.) and large-leaved linden (*T. platyphyllos* Scop.) are native to most European countries, with a distribution range extending from southern Finland to southern Italy and the Caucasus. The natural distribution range of *T. cordata* is much wider, but it is widespread in Central and Eastern Europe. The natural presence of *Tilia* cordata extends to southern Norway and Finland in the north and up to 1500 m in the central Alps. *T. platyphyllos* has a smaller range that extends a little further south, but the northern half extends only to southern Sweden, with the eastern limit ending in central Europe [3,4]. These two linden species may naturally hybridize, and the hybrids are identified taxonomically as European linden (*Tilia* × *europaea* L., syn. *T.* × *intermedia* DC., and *T*. × *vulgaris* Hayne). Due to its longevity and ability to withstand pruning, *T*. × *europaea* were very common in the parks and gardens of Central and Northern Europe in the 17th and 18th centuries [5,6,7,8]. 

*T. cordata* was abundant in the forests around the Baltic Sea in the late Holocene until 3000 BC. A colder climate since the mid-Holocene is responsible for the decline in linden due to less favorable climate for seed germination and bee presence during pollination [9]. The decline in lindens may also have been due to farming in forest areas when large forests were cut down and forest management was carried out, for example, by afforestation or pollarding for fodder [10]. The subsequent cooling of the climate and the development of agriculture gradually reduced linden forests in Europe, in some places completely displacing them, in others leaving small and fragmented populations. The pollination, fruiting, and seed germination of linden trees are markedly reduced by unfavorably cool or even cold temperatures [11], which may have reduced their competitiveness with other forest tree species. 

After the cooling following the warm Atlantic period in the Baltic countries and Poland, linden trees have survived in small, fragmented populations in forests and as single trees in homesteads, estates, and town parks [12,13,14,15,16,17,18]. Currently, as the climate is warming, favorable conditions for spreading linden back into forests occur both from native forest populations and from domesticated groups [12]. *T. platyphyllos* and *T.* × *europaea* are exotic species in Lithuania, mostly growing as decorative trees in parks and other urban places [13,19]. Lindens, as ornamental trees in streets and parks, remain among the most planted species in European cities [20]. Lindens are among the most resistant trees to water shortage, drought, pollution, and pruning [21]. The origin of *T. platyphyllos* and *T*. × *europaea* in Lithuania is unknown. The likely sources of the spreading of the exotic *Tilia* species are botanical gardens of neighboring countries. In such a way, the exotic *Tilia* species gradually spread to the territories of monasteries, cities, and towns of Lithuania. The few available genetic studies on the evolutionary origin of the naturally growing *T. cordata* populations in Lithuania revealed that the gene pool of these native trees has remained diverse, and the warming climate is favorable for the growth of linden trees [12]. 

The morphological identification of different linden species is complex, especially when the trees are young. Also, identifying species of adult trees is a complex issue, especially when hybrids combine parental traits, or when introgression occurs [1,7,22,23,24,25,26]. The use of dimensional characteristics (leaf length, width, etc.) unfortunately did not help distinguish linden species, especially the hybrids [27]. Meanwhile, Andrianjara et al. [28], who performed a comparison of morphological and genetic analysis, stated that morphological characteristics are an important tool for linden species identification. 

Chemotaxonomic and genetic markers can also be used to identify linden trees. A comprehensive validated ultra-high performance liquid chromatography (UHPLC) coupled with the diode array detection (DAD) mass spectrometry method (uhplc-dad-ms/ms) was used to distinguish the five most important *Tilia* species in Europe based on phytochemical analyses of extracts prepared from their flowers [29]. However, in this case, *T. cordata* was distinguished from *T. platyphyllos*, and *T*. × *europaea* overlapped with *T. platyphyllos*. Different molecular analyses showed differentiation between both species and their naturally occurring hybrids (*T*. × *europaea*) is easily distinguished [23,26,28,30]. Genetic studies can be useful for species genetic profiling of trees, identifying existing clones in an area, and assessing potential disease risk [28].

In this study, we compared the morphological characteristics of old-growth *T*. × *europaea* trees, which grew in the Pazaislis monastery for about 150 years and were identified by the DNA microsatellite method as the hybrids between *T. cordata* and *T. platyphyllos*, with the other linden species growing around the monastery. As a control, we studied samples of *T. cordata* and *T. platyphyllos* trees growing in a park near the Pazaislis monastery. Our study will expand the knowledge regarding the possibility of using morphometric markers as a simple and quick method for linden hybrid identification. The main objective of our study is to find simple and effective morphometric markers for the identification of *T. cordata* and *T. platyphyllos* hybrids.

## 2. Results and Discussion

Based on the key descriptor for *Tilia* species (Appendix A) and the morphometric analysis of leaves and bracts, our specimens could be divided into four morpho-groups that correspond to four *Tilia* species, including the two hybrid species (Table 1, Table 2, and Appendix A). When identifying the *Tilia* species, the morphometric traits of leaves and nut morphology traits were the most important. Note, however, that our results primarily apply to northerly Europe, whereas elsewhere, drastically different adaptive environments may lead to deviations in the morphometric trait values reported in our study. 

In the sampling site of the Pazaislis monastery, the morphometry-based taxonomy of *Tilia* individuals was as follows (Table 1, Table 2, and Appendix A): (a) the six old-growth trees from the front yard of the Pazaislis monastery (that, based on the DNA markers, were identified as *T*. × *europaea*) were identified as *T*. × europaea; outside the monastery yard no *T.* × *europaea* was found among the sampled *Tilia* sp. trees based on our morphometric descriptor (Figure 1), (b) the four old-growth trees of *T*. × *europaea* var. *europaea* ‘Pallida’ growing immediately outside the monastery fence (Figure 1) and (c) the remaining 15 old growth trees were identified as *T. cordata* (Figure 1). In Girionys park, the sampled *Tilia* species were morphometrically identified as (a) *T*. *platyphyllos,* the three trees that were assigned as *T*. *platyphyllos* by the DNA markers; and (b) *T. cordata*, all the remaining trees studied in Girionys park. 

Morphologically, the *T*. × *europaea* var. *europaea* ‘Pallida’ individuals strongly differentiated from the remaining *Tilia* species by the obliquely truncated leaf base (unlike other *Tilia* species with cordate leaves) and the presence of epicormic shoots on the stems. 

The *T. platyphyllos* individuals from Girionys park could morphometrically be assigned to *T. platyphyllos* subsp. *cardifolia*. The *T. platyphyllos* individuals were easily distinguished by white, bristly, long, visible hairs on current year’s shoots, leaves, and even petioles. The upper side of their leaves was slightly hairy at the veins, while the other species had no hairs on the upper side of the leaves (Table 1, section A). The leaf margins of *T. platyphyllos* were ciliated (Table 1, section B5). 

In the *T*. × *europaea* individuals from the Pazaislis monastery front yard, the marginal leaf teeth were intermediate between *T. cordata* and *T. platyphyllos*. The latter finding agrees well with [28], who described in their study that *T. cordata* and *T.* × *europaea* are difficult to distinguish based on the marginal leaf teeth. The *T. platyphyllos* individuals were distinguished by sharply pointed teeth. On the underside of the leaves, *T. cordata* was outstanding in its bluish-green color and brownish hairs at the base of the lamina and in the branches of the veins (Table 1, section B). In *T*. × *europaea* and *T*. *platyphyllos*, the lower side of the leaves was green, and the hairs were lighter, but *T*. *platyphyllos* had hairs not only in the branches of the veins but also in the veins. The third-row veins of *T*. × *europaea* were prominent and horizontal like in *T platyphyllos*. This could be one of the most striking morphometric markers distinguishing *T. cordata* from *T.* × *europaea* (Table 1, section B4). 

The petiole of *T. platyphyllos* differs from the other *Tilia* species by its hairiness (Table 1, section C). Last-year twigs differed markedly among all the *Tilia* species (Table 1, section D). The twigs of *T. platyphyllos* were green, with stellate hairs, the stomas slightly raised, and those for *T. cordata* were pink, glabrous, with narrow stomas. Meanwhile, *T*. × *europaea* twigs were intermediate in color, with wider stomas than *T. cordata*, and the stomas were not raised. The buds of *T.* × *europaea* are more like those of *T. platyphyllos* (Table 1, section E). 

The hairiness of the bracts could be another morphometric marker well discriminating among the *Tilia* species. The bracts of *T. cordata* and *T.* × *europaea* var. *europaea* ‘Pallida’ were glabrous, *T*. × *europaea* and *T. platyphyllos* had hairs at the base of the pedicle, and *T. platyphyllos* contained hairs on the lower side of the bracts on the midvein. 

The *T. platyphyllos* individuals contained the largest flowers with bright yellow petioles, sharp sepals, as well as large pistils and stamens (Table 1, section G), whereas the petioles of the other *Tilia* species studied were light yellow in color. Flowers of the *T*. × *europaea* var. *europaea* ‘Pallida’ individuals were the smallest. The *T*. × *europaea* individuals are distinguished by narrower flowers than those of *T. cordata* and by intermediately sized pistils if compared with the parental species.

Hardiness, ribs, and the size of the nuts can also be used to identify *Tilia* species (Table 1, section H). Nuts of *T. platyphyllos* were the largest in diameter, hard to crush, spheroid to broadly obovoid, with prominent ribs. Nuts of *T. cordata* were the smallest, spheroid or obovoid, without prominent ribs, and nuts of *T*. × *europaea* have average parameters. Meanwhile, the nuts of *T.* × *europaea* var. *europaea* ‘Pallida’ are more like those of *T. cordata* with the more prominent ribs.

The ANOVA analysis revealed highly significant differences among the *Tilia* species in all the morphometric traits (Table 3). According to the Tukey LSD test, *T. cordata* exhibited the strongest differentiation from the other *Tilia* species (Table 3). Additionally, *T.* × *europaea* var. *europaea* ‘Pallida’ displayed a significant difference from the other *Tilia* species, specifically by containing the largest bract PL. The statistical differences between the *Tilia* species in the leaf traits were stronger than in the other morphometric traits measured. All the *Tilia* species were significantly different in the BL, AL, and LA25 variables. Most of the morphometric traits measured (LWA, BL, PMP, MPW, LWA/PL, and LWA/MPW) possessed higher values for *T. platyphyllos* and *T*. × *europaea* var. *europaea* ‘Pallida’ species. *T. platyphyllos* differs from the rest by low AL. Meanwhile, *T*. × *europaea* morphometric traits were intermediate or alike to *T. cordata*.

The PCA analysis showed that the first two principal components explained 71% of the total variation in the morphometric traits. The first principal component is strongly positively and negatively associated with BL, LWA, PMP, LWA_MPW, LA10, and LA25 traits, respectively. The second principal component mainly represents the variation in MPW and, to a lesser extent, the variation in LA25, PL, and LA10 traits (Figure 2). The traits MPW and PL were of the utmost importance for differentiating *T. platyphyllos*, while the traits LWA_MPW and LA25 LA10 were crucial for distinguishing *T*. × *europaea* var. *europaea* ‘Pallida’ individuals. The PCA plots indicate that species *T. cordata* and *T*. × *europaea* cluster into a single group, while the *T. platyphyllos* and *T*. × *europaea* var. *europaea* ‘Pallida’ individuals cluster into separate groups (Figure 2). The following morphometric traits were important for differentiating these PCA species groups: LWA, BL, and LWA_MPW (Appendix A); LA1, AL, and LA2 (Appendix A); PMP, MPW, and LWA_PL (Appendix A); LA1, PL, and PMP (Appendix A); and BL, LA2, and AL (Appendix A).

The local environment can have a significant effect on the growth of trees, their response to environmental conditions, and genetic variability within a species [7,28,31,32]. For example, in the color of twigs and leaves, the shape of leaves can change depending on the light regime. Therefore, for morphological identification, tree samples must be collected simultaneously and from similar solar exposure in the same section of the crown, and *Tilia* species taxonomy must rely on more markers than the color of the twigs alone. Here, DNA markers are very helpful, as in our case, a clear molecular separation of *T*. *cordata* and *T. platyphyllos* at the Tc8 microsatellite locus. Consequently, the hybrid *T.* × *europaea* must be a homozygote at the Tc8 locus containing a species-specific allele from each of the parent species. In agreement with the molecular identification, the *T*. *europaea* individuals exhibited intermediate/parental morphology in the color of the twigs of the first year, the color of the petiole, the size of the nuts, and their hardiness. However, not all the morphological and morphometric characteristics of the *T*. × *europaea* shoots were intermediate between the parental species. The following shoot morphometry of the *T*. × *europaea* was closer to the *T*. *platyphyllos*: the color of the leaf lower side, the hair color, the horizontality and raise of the tertiary veins, the number of bud scales, and the hairiness of the bract at the base. Meanwhile, the following morphology traits of *T*. × *europaea* were more similar to *T. cordata*: the shape and size of the leaf, the hairless upper side of the leaf, the hairiness of the lower side of the leaf, the shape of the marginal teeth, and the hairless petiole. Pigott [33] described the tertiary veins as moderately raised in hybrids, but in our case, they were raised, and this was confirmed by Ramanauskas [34]. Since our samples were collected in August, we could not assess the hairiness of leaf veins, petiole, and twigs, which Piggot [33] estimated as intermediate since some of the hairs had already fallen by this time.

**Figure 3 plants-12-03567-f003:**
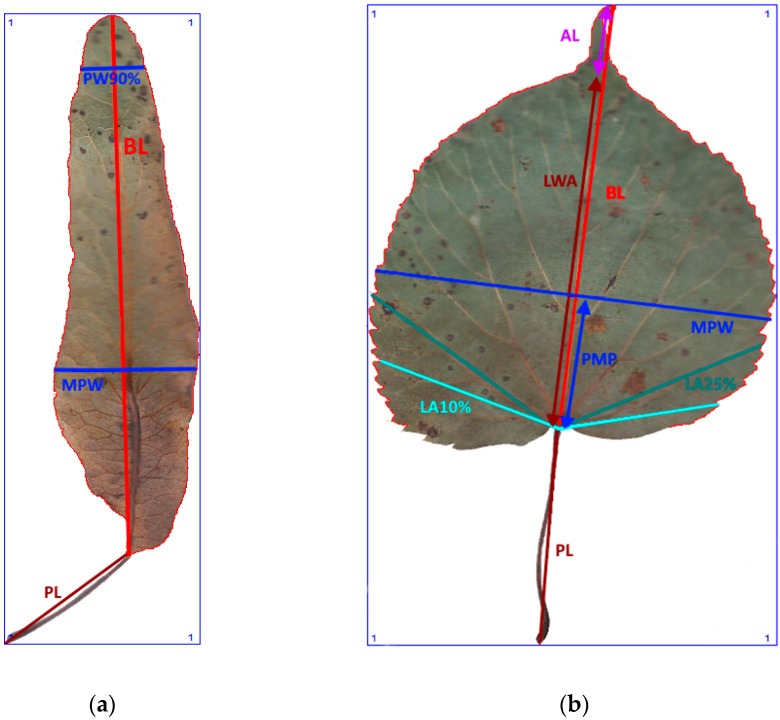
The morphometric traits of bracts (**a**) and leaves (**b**) of *Tilia* sp. measured with WinFolia (2016).

Meanwhile, the *T. europaea* var. *europaea* ‘Pallida’ was like the *T. cordata* in the following traits: nut shape and hardness, non-hairy bracts, non-hairy twigs and petioles, and non-hairy veins. The marginal teeth of the leaves were similar to those of the *T. platyphyllos*. It was distinguished by a truncated leaf base, little or no hairs on the branching veins, and a dark red petiole. When evaluating the leaf morphometry, the traits of *T. cordata* and *T*. × *europaea* overlapped. Meanwhile, *T*. × *europaea* var. *europaea* ‘Pallida’ and *T. platyphyllos* differed from these LWA_MPW and PMP. *T.* × *europaea* var. *europaea* ‘Pallida’ differed from the other limes in L A10 and AL, while the AL of the *T. platyphyllos* was the lowest. These traits can be used together with morphological ones to distinguish linden tree species.

Based on DNA markers, the *T.* × *europaea* growing in the Pazaislis monastery were identified as copies of a single individual (clones). It is likely that they were grafted. However, no morphologically similar lindens were found in the park around this monastery. Indicating that those six *T*. × *europaea* individuals were planted on purpose. Most of the lindens growing around the monastery are *T. cordata*. Often, the problem in cities and parks is that the trees planted for aesthetic purposes are trees of a single clone, as is the case in the Pazaislis monastery. This reduces the genetic diversity of urban trees and makes the trees less resistant to diseases and pests [28,35].

## 3. Materials and Methods

### 3.1. DNA Study

While carrying out a country-wide genetic study on 543 adult *T. cordata* trees in Lithuania based on nuclear microsatellite markers [12], we identified a nuclear microsatellite locus Tc8 unambiguously discriminating between *T. cordata*, *T. platyphyllos*. The set of nuclear microsatellite loci was developed for genetic studies of *T. cordata* by [12,22]. Danusevicius et al. [13] found that the microsatellite locus Tc8 was fixed to a 140 bp allele in *T. cordata* (541 sampled individuals) and amplified alleles above 160 bp in two trees with *T. platyphyllos*-like morphology (sampled in a national park). In the present study, our intention was to use this Tc8 locus to investigate the taxonomy of six old-growth *Tilia* trees growing in the Pazaislis monastery yard (Kaunas, central Lithuania N 54°52′37.00″ E 24°1′13.54″). Each of these six trees had a disputed taxonomy, with uncertainty regarding whether they belonged to *T. cordata* or *T. platyphyllos*. 

To verify the nuclear microsatellite Tc8 locus as an efficient species-specific marker, we used the Tc8 microsatellite locus to genotype further four *T. platyphyllos* individuals from Girionys park (Kaunas, central Lithuania) located ca. 10 km away from the old-growth *Tilia* trees in the Pazaislis monastery. Additionally, we used a set of 14 nuclear microsatellite loci to study genetic associations between the six old growth *Tilia cordata* trees from the front yard of the Pazaislis monastery (the loci are further described in [12] and the references therein). The DNA extraction, PCR, and capillary electrophoresis procedures are described in detail [12]. Briefly, the DNA was extracted from fresh leaves according to a modified ATMAB protocol. The PCR was run on Applied Biosystems Thermo Cycler GeneAmp PCA System 9700 (Applied Biosystems, Foster City, CA, USA) as follows: initial denaturation step at 95 °C for 15 min, followed by 25 cycles each of 94 °C for 30 s, annealing temperature at 54 °C for 1 min, 30 s, and extension at 72 °C for 30 s, followed by the final extension step at 60 °C for 30 min.; the PCR products were separated by capillary electrophoresis on ABI PRISM™ 310 genetic analyzer (Lincoln Centre Drive, Foster City, CA 94404 USA) and the alleles were scored on GENEMAPPER soft. Ver. 4.1. The microsatellite genotyping revealed that all four *T. platyphyllos* trees contained alleles above 160 bp at the Tc8 microsatellite locus. We concluded that the nuclear microsatellite locus Tc8 is an efficient DNA marker for discrimination between *T. cordata* and *T. platyphyllos*. We subsequently genotyped the six old-growth individuals from the Pazaislis monastery with mixed *T. cordata* × *platyphyllos* morphology by using a set of 14 genomic microsatellite markers earlier used by [12] and developed by [22] (including the Tc8 locus). All four individuals of *T. platyphyllos* from Girionys park that were genotyped, and the six-old growth *T. cordata* × *platyphyllos* hybrids from the Pazaislis monastery were later selected for the morphometric evaluation. 

### 3.2. Morphometric Study

For the morphometric study in September 2022, twigs with fruits and leaves and at the end of June 2023, twigs with flowers were collected from 31 mature *Tilia* spp. Trees growing in two parks located in Kaunas city, central Lithuania: (a) six old-growth *T*. × *europaea* trees in the Pazaislis monastery front yard (the same individuals as were genotyped and based on Tc8 locus assigned as *T.* × *europaea*), 16 old-growth trees of *T. cordata* (not genotyped), and 4 old-growth trees of *T. europaea* var. *europaea* ‘Pallida’ (not genotyped), all growing in the Pazaislis monastery park (N 54°52′37.00″ E 24°1′13.54″); and (b) 3 mature trees of *T. platyphyllos* (that were genotyped and based on Tc8 locus assigned to *T. platyphyllos*) and 2 mature trees of *T. cordata* in Girionys park near Kaunas (not genotyped) (Figure 1). The leaves were sampled within a few day intervals to avoid deviations due to color variation of the leaves and twigs [28].

To prepare samples, two sun-exposed twigs with fruits were collected from the lower part of the crown (2–5 m above the ground). The best-preserved twigs with leaves and fruits were selected and dried by pressing samples within folded newsprints for several days. All the collected specimens were assigned to the correct *Tilia* species by using the morphology identification key of De Langhe [36]. This key is based solely on vegetative characteristics. We also considered a monograph by Pigott [7] (2012) describing the morphology of all vegetative and reproductive structures of twigs. We also used morphological descriptions of linden species (*Tilia* L. spp.) that were described by [34,37] in our assessments. Based on the descriptors of the above-mentioned authors, we constructed and used a key for the morphological identification of *Tilia* species (Appendix A). The morphological identification of linden trees was based on the combination of several criteria. We paid attention to the color of the leaf’s upper side, the leaf’s form, the teeth of the leaf’s margin, the color of petiole and twigs, and determined the number of pairs of secondary veins of leaves, the number of bud scales, the length of the upper scale (more or less than half the length of the bud), hardness, and the shape of nuts. A 400× Series Digital Microscope (EduScience, London, UK) was used to evaluate the hairs on the surface of the leaves, petioles, bracts, and twigs.

### 3.3. Morphometric Measurements

Leaf morphometric traits of 31 linden trees with 10 to 12 leaves and 3–5 bracts per tree were scanned and assessed: *T. cordata* (18 trees), *T*. × *europaea* (6 trees), *T*. × *europaea* var. *europaea* ‘Pallida’ (4 trees) and *T. platyphyllos* (3 trees, all genotyped). The leaf mean values of the morphometric traits were used as units of observation in the data analysis. In the morphometric investigation, we chose 8 leaf and 4 bract traits for identifying the linden species. The WinFolia 2016 Leaf analyzer program, Basic version (Regent Instruments Inc., Quebec, QC, Canada) was used to score these traits (Table 4, Figure 3). We also used the following computed variables for bract and leave morphometry: ratio BL/PL for bracts and ratio LWA/MPW and LWA/PL for leaves (Table 4).

We used the SAS 9.4 software, using the PROC MEANS procedure for the descriptive statistic (mean, standard deviation). The PROC GLM procedure with Tukey’s least significant difference test was used for the one-way analysis of variance (ANOVA) to determine the species effect on the bract and leaf morphometric traits. We used PROC PRINCOMP, PROC CORR, and PROC SGPLOT procedures for the calculation and visualization of Principal Component Analysis (PCA) to evaluate which morphometric traits best correspond to the different species of the linden genus [38].

Nuts of the *Tilia* species were compared on the basis of their hardiness and presence, as well as the sharpness of the nut ribs (Appendix A). For flowers, we scored the color of the petioles and shape, as well as the size of the flower structures. 

## 4. Conclusions

The morphological key for distinguishing *T*. × *europaea* individuals from their parental species of *T. cordata* and *T. platyphyllos* is as follows: intermediate leaf size between the two parental species; the purely green color of the lower side of the leaves (lower side of leaves of *T. cordata* is bluish green); surface leaf hairiness is similar to that of *T. cordata,* but the hair at the vein basis at the upper side of the leaves are much lighter in color (*T. cordata* contains brownish hair at the vein basis); the third-row veins are bold and horizontal (in contrast to *T. cordata*); hairless leaf petiole (haired leaf petiole only in *T. platyphyllos*); terminal buds contain three scales and are smaller than buds of *T. platyphyllos.* There is a small patch of simple hairs in the axil of the peduncle, and this way, it differs from that of *T. cordata*; the hardiness and ribs of the nuts are intermediate between the parental species, and there are relatively lower values for morphometric traits of LWA, BL, and LWA_MPW.

## Figures and Tables

**Figure 1 plants-12-03567-f001:**
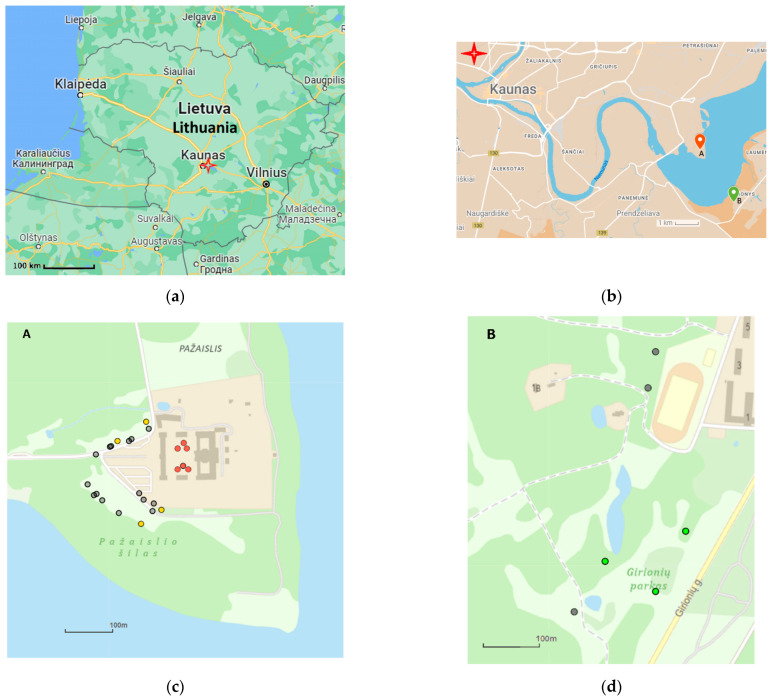
Location of the *Tilia* sp. individuals sampled in the two parks in central Lithuania: (**a**) location within Lithuania, (**b**) location in the city of Kaunas, (**c**) the samples in the Pazaislis park, (**d**) the samples in Girionys park. Species coded by cycle color (morphologically assigned): grey—*Tilia cordata* Mill.; green—*T. platyphyllos* Scop.; red—*T*. × *europaea* L.; yellow—*T*. × *europaea* var. *europaea* ‘Pallida’.

**Figure 2 plants-12-03567-f002:**
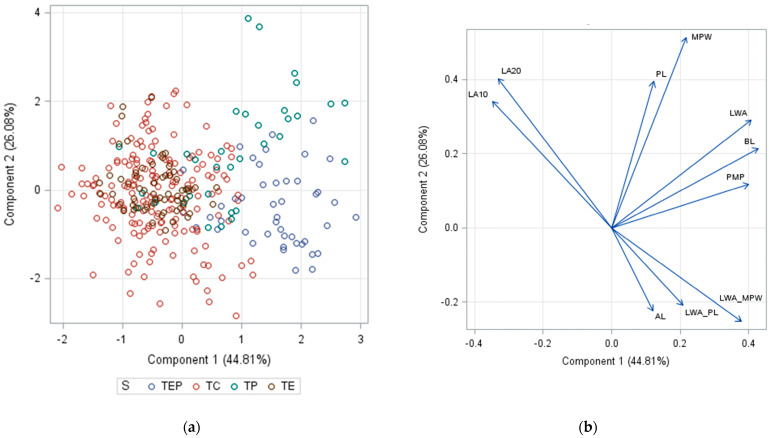
Results of the principal component analysis (PCA) on the morphometric traits of *Tilia* leaves: (**a**) scatter plot of individual observations on *Tilia* specimens based on the two first principal components, (**b**) description of the principal components (Wicklin 2019). The abbreviations of *Tilia* species (S) are explained in Table 1, and the abbreviations of the traits are in Table 4.

**Table 1 plants-12-03567-t001:** Comparison of the morphometric traits of the three *Tilia* species identified in the samples collected in the Pazaislis monastery park and Girionys park.

*Tilia cordata* (TC)	*Tilia* × *europaea* (TE)	*Tilia* × *europaea* var. *europaea* ‘Pallida’ (TEP)	*Tilia platyphyllos* (TP)
A. Upper side of leaf
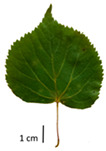	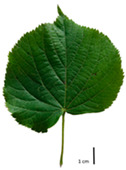	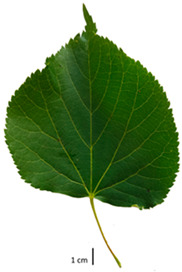	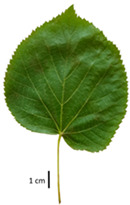
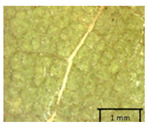	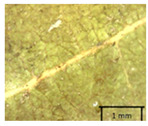	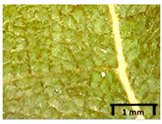	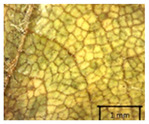
B. Lower side of leaf
B1. Whole leaf
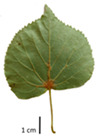	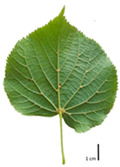	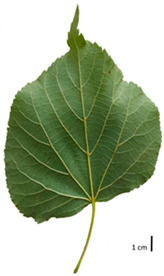	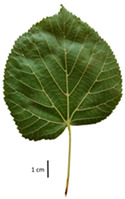
B2. Base of main veins
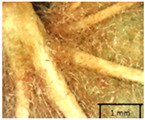	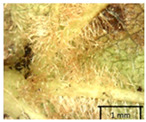	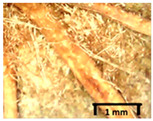	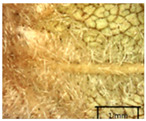
B3. Branches of veins
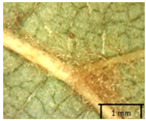	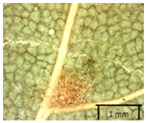	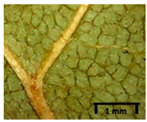	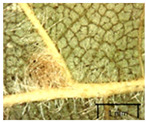
B4. Third-row veins
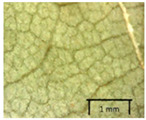	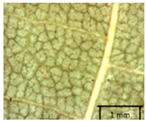	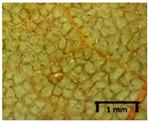	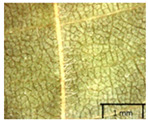
B5. Leaf margin
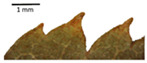	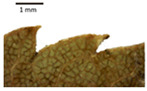	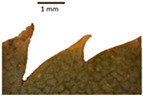	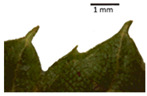
C. Petiole
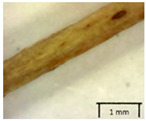	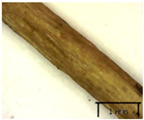	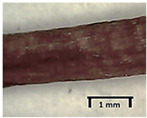	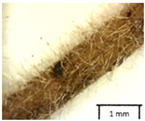
D. Twigs
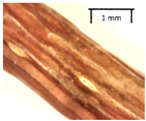	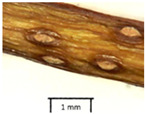	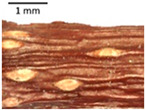	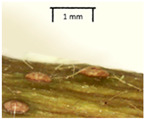
E. Buds
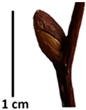	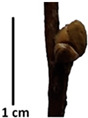	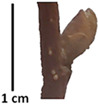	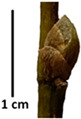
F.Bract
F1. Lower side
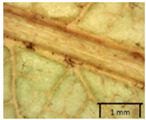	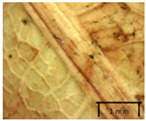	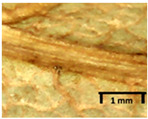	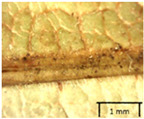
F2. Upper side
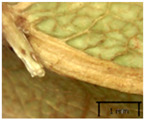	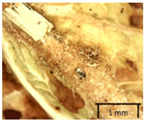	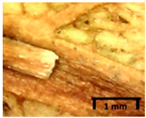	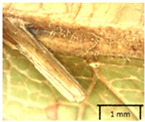
G. Flowers
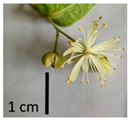	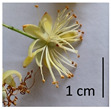	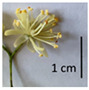	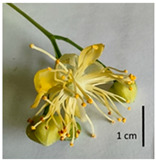
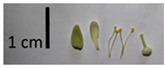	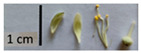	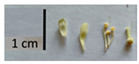	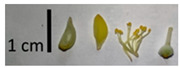
H. Nuts
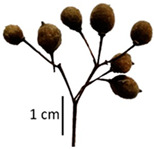	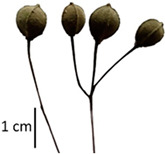	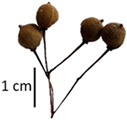	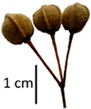

**Table 2 plants-12-03567-t002:** The main values of the main morphological traits of *Tilia* species from the samples collected in the Pazaislis monastery park and Girionys park (TC—*Tilia cordata*, TE—*T*. × *europaea*, TEP—*T*. × *europaea* var. *europaea* ’Pallida’, TP—*T. platyphyllos*).

No	Species	TC	TE	TEP	TP
1	Nut form, fragility, ribs	Obovoid/spheroid, fragile, blurred	Spheroid/ellipsoid, average hard, average bright	Spheroid/obovoid, fragile, slightly bright	Spheroid/obovoid, hard, bright
2	Bract: petiole on lower/upper side, Length (cm)	−/−, 6–7	−/+, 8–9	−/−, 8–9	±/+, 7–8
3	Leaf form: Width × length (cm); base	6.5 × 5.5, cordate	6.6 × 5.5, cordate	7 × 7, semicordate	8 × 7.5, shallowly to deeply cordate
4	Veins: third row (raised/horizontal or no), secondary (number of pairs)	−, 5–6	+, 7–8	−/+, 6–7	+, 7–9
5	Number of bud scales, Size of the outer scale (part from bud)	2 1/2	3 <1/2	2–3 ≥1/2	3, ≤1/2
6	Upper part of leaf hairiness	No	No	No	Yes, at base (rare), veins and petiole (dense)
7	Lower part of leaf color, hairiness	Pale green; brown stellate hairs in veins axils	Green; light brown stellate hairs in veins axils	Pale green; light brown stellate hairs in base and some veins axils	Green; white, dense fasciculate in axils, and medium density simple hairs in secondary and third row veins, and on petiole
8	Marginal teeth	Subacute	Subacute	Apiculate/subacute	Apiculate

**Table 3 plants-12-03567-t003:** Descriptive statistics (range, mean, and standard devotion) and results of the ANOVA on the species effect on the morphometric traits, as well as the result of the Turkey LSD test, where the different letters show significant differences in morphometric traits between the *Tilia* species sampled in the two Lithuania parks.

Morphometric Traits *, cm	TC **	TE	TEP	TP	R^2^	F Value	*p*
	Min–Max Mean ± SD (*n* − 1) Turkey Index						
Bracts							
Total number of measured bracts	85	30	20	14			
BL	2.79–10.10	7.24–10.19	7.37–10.48	5.56–9.72	0.26	16.55	<0.0001
	6.85 ± 1.59	8.48 ± 0.85	8.56 ± 7.37	7.98 ± 1.05			
	b	a	a	a			
BL_PL	0.86–27.42	0.89–11.97	1.82–5.74	3.98–28.22	0.24	15.07	<0.0001
	5.19 ± 3.94	6.59 ± 2.19	3.41 ± 1.10	11.78 ± 7.18			
	b	b	c	a			
PW90	0.28–1.93	1.04–1.88	0.99–2.07	0.89–1.70	0.18	10.47	<0.0001
	1.14 ± 0.32	1.45 ± 0.17	1.39 ± 0.30	1.30 ± 0.27			
	b	a	a	ab			
PL	0.19–6.86	0.74–9.07	1.63–4.64	0.27–1.40	0.17	10.23	<0.0001
	1.76 ± 0.93	1.58 ± 1.44	2.75 ± 0.86	0.87 ± 0.37			
	b	bc	a	c			
MPW							
Leaves							
Total number of measured leaves	180	60	45	35			
LWA	3.24–7.31	4.60–6.71	5.75–9.42	5.49–10.05	0.51	109.73	<0.0001
	5.52 ± 0.78	5.53 ± 0.43	7.31 ± 0.79	7.58 ± 1.36			
	b	b	a	a			
BL	4.53–9.15	5.45–7.91	6.91–10.49	6.19–10.81	0.46	90.84	<0.0001
	6.81 ± 0.89	6.69 ± 0.53	8.88 ± 0.87	8.34 ± 1.30			
	c	d	a	b			
LWA_MPW	0.71–1.09	0.75–0.95	0.86–1.26	0.83–1.11	0.45	87.89	<0.0001
	0.86 ± 0.08	0.84 ± 0.05	1.03 ± 0.09	0.97 ± 0.06			
	c	c	a	b			
LA10	132–158	147–157	128–152	144–157	0.36	60.13	<0.0001
	151.28 ± 4.5	152.23 ± 2.27	142.24 ± 5.8	149.31 ± 3.45			
	ab	a	c	b			
AL	0.43–2.11	0.61–1.51	0.36–2.59	0.23–1.31	0.35	56.51	<0.0001
	1.30 ± 0.29	1.16 ± 0.18	1.57 ± 0.38	0.76 ± 0.28			
	b	c	a	d			
PMP	1.05–3.24	1.65–3.01	1.65–3.75	1.32–4.16	0.34	53.69	<0.0001
	1.96 ± 0.43	2.19 ± 0.31	2.85 ± 0.48	2.66 ± 0.80			
	c	b	a	a			
LA25	107–132	118–131	104–123	115–127	0.33	51.74	<0.0001
	122.29 ± 5.4	124.32 ± 3.12	113.42 ± 4.88	120.91 ± 3.09			
	c	a	d	b			
MPW	3.85–9.44	5.53–7.73	5.87–9.57	5.86–10.04	0.16	20.29	<0.0001
	6.45 ± 1.05	6.63 ± 0.52	7.12 ± 0.98	7.79 ± 1.30			
	c	cb	b	a			
LWA_PL	0.60–2.17	0.67–2.12	0.77–2.28	0.73–2.47	0.10	12.24	<0.0001
	1.45 ± 0.27	1.57 ± 0.25	1.67 ± 0.23	1.63 ± 0.32			
	b	a	a	a			
PL	2.24–9.42	2.52–8.46	3.18–9.58	3.32–11.92	0.09	10.28	<0.0001
	3.98 ± 1.13	3.66 ± 1.06	4.45 ± 0.92	4.94 ± 1.93			
	bc	c	ab	a			

* Description of morphometric traits shown in Table 4. ** Description of linden species shown in Table 1.

**Table 4 plants-12-03567-t004:** List of morphometric traits of leaves and bracts measured for the studied trees of *Tilia* sp. (depicted in Figure 3).

No.	Morphometric Traits	Abbreviation
1	Blade length, cm	BL
2	Apex length, cm	AL
3	Blade length without apex, cm	LWA
4	Maximum blade width, measured perpendicular to blade, cm	MPW
5	Length to position where maximum blade width, cm	PMP
6	Blade width perpendicular to blade length at 90% blade length, cm	PW90
7	The Blade Lobe Angle at 10% Blade Length, degree	LA10
8	The Blade Lobe Angle at 25% Blade Length, degree	LA25
9	The Petiole Length, cm	PL
Derived variables		
10	Ratio (Blade length and petiole length) (1/9)	BL_PL
11	Ratio (Blade length without apex and Maximum blade width) (3/4)	LWA_MPW
12	Ratio (Blade length without apex and Petiole Length (3/9)	LWA_PL

## Data Availability

The data presented in this study are available within the article and in Appendix A.

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
