# Peer review of "Dendrological Secrets of the Pazaislis Monastery in Central Lithuania: DNA Markers and Morphology Reveal Tilia × europaea L. Hybrids of an Impressive Age"

_plants, 2023, doi:10.3390/plants12203567_

Round 1

Reviewer 1 Report

Jurkšienė et al. investigate hybrids between Tilia cordata and T. platyphyllos in Lithuania using morphological data and nuclear microsatellites. Their results are also interesting for general investigation of hybridization between the species. The results are presented in much detail and well discussed.

The manuscript just needs to be re-worked by a native speaker to correct some spelling mistakes, e.g. on line 12 "that " instead of "the" and line 136 "remaining" instead of "reaming"; also some phrases sound strange, e.g. line 43 "taste of honey".

Author Response

Reviewer coment

Answer

Comments and Suggestions for Authors Jurkšienė et al. investigate hybrids between Tilia cordata and T. platyphyllos in Lithuania using morphological data and nuclear microsatellites. Their results are also interesting for general investigation of hybridization between the species. The results are presented in much detail and well discussed.

Thank you for your very encouraging comments on our manuscript.

Comments on the Quality of English Language The manuscript just needs to be re-worked by a native speaker to correct some spelling mistakes, e.g. on line 12 "that " instead of "the" and line 136 "remaining" instead of "reaming"; also some phrases sound strange, e.g. line 43 "taste of honey"

English was revised by a native speaker

Reviewer 2 Report

G. Jurksiene and coworkers describe a morphometric analysis of the most common NW European lime tree species. Through the combination of earlier developed indicative SSR markers with morphometry they were able to identify the characters most useful to discern the otherwise often hard to determine species. 

The study is well done and could help to increase knowledge about the occurrences of the specific species and their hybrid. Thus, it was an interesting reading for me. 

The only point of concern I have is the question of how well the described morphological traits might work in other areas of Europe to discern the species. For this it would be nice to have other individuals from further south in Europe included in the study. However, I understand that this is now not feasible and hope that other botanist might use the study to check out their local populations to see if the differences are present also there.

The study is clearly presented and the methods are all appropriate. My only criticism regards the sloppy formatting of mostly the taxonomic names (highlighted in the PDF), some sentences where English seems to be not really comprehensive (also highlighted in the text), and references, which are a mess (I started to mark up the inconsistencies but stoped in between – it should be clear to the authors that some work is needed to stick to the journal's formatting and make refs consistent).

See before

Author Response

Reviewer coment

Answer

G. Jurksiene and coworkers describe a morphometric analysis of the most common NW European lime tree species. Through the combination of earlier developed indicative SSR markers with morphometry they were able to identify the characters most useful to discern the otherwise often hard to determine species. The study is well done and could help to increase knowledge about the occurrences of the specific species and their hybrid. Thus, it was an interesting reading for me.

Thank you for your encouraging comments on our manuscript.

The only point of concern I have is the question of how well the described morphological traits might work in other areas of Europe to discern the species. For this it would be nice to have other individuals from further south in Europe included in the study. However, I understand that this is now not feasible and hope that other botanist might use the study to check out their local populations to see if the differences are present also there.

Yes, this a is a good point. We have added a comment on that in the discussion. We added the following sentence in the 1st paragraph of the Results and Discussion section:

 Note, however, that our results primarily apply to northerly Europe, whereas elsewhere drastically different adaptive environments may lead to deviations in the morphometric trait values reported in our study.“

The study is clearly presented and the methods are all appropriate. My only criticism regards the sloppy formatting of mostly the taxonomic names (highlighted in the PDF), some sentences where English seems to be not really comprehensive (also highlighted in the text), and references, which are a mess (I started to mark up the inconsistencies but stoped in between – it should be clear to the authors that some work is needed to stick to the journal's formatting and make refs consistent).

Thanks for the comments. We corrected the taxonomic names as requested and reviewed the references.

Reviewer 3 Report

The work presented is solid but based on a scarce number of individuals. Which is understandable given the ancestry and rarity of the specimens. 

However, the method used is NOT developed by the authors for the first time in this paper as the narrative and abstract seems to indicate. Microsatellite locus T8 was not detected by the authors in this paper, it was previously found and reported in other papers,  so they are using a method that exists already. The narrative has to reflect this clearly. They are using an existing method. Not a novel method.

The way it is written seems to indicate you developed it and present it for the first time in this paper  which is incorrect. Danusevicius (12-13) did this before you and called it Tc8. I am not sure why you use T8 here and then refer Tc8 for the same test by Danusevicius (12-13). If it is the same loci you should use the same nomenclature, Tc8.  It doesn't matter he is an author in the current paper. The method already exists. It is not novel in this paper.

 It is difficult to know the number of replicates for each measurement or molecular test as they are not listed across the entire paper. Authors should tell the reader to check Table 3 or list (N=xxx) when speaking about measurements or genetic profiles. We need to know how many times something was done to give statistical support.

 Part of the problem to understand the paper is that it is very succinct and mostly refers to methodologies of the authors former paper for the genetics. The genetic methods are described in half a line “The DNA extraction, PCR and capillary electrophoresis procedures  are described in [12]. Nothing else. Not very clear I would say. At least write :briefly, ….” and describe some of the methods. Also, what happened to the other 13 microsatellite listed in the abstract? They don’t appear anywhere in the paper. I assume this is what you call “genotyped”? and where are the methods for this? Are the 13 microsatellites across the entire genome? this is important information. And where is this T8 located??? nuclear, mitochondria? I assume nuclear but nothing is mentioned in the text. Depending on this the results are very different if it is mitochondria. That would be heteroplasmy, not an hybrid. This is crucial information.

It is simply too succinct and we don’t know how the molecular part of the work was done. The morphometry is far better described.

 Please note the word THE before the monastery name is necessary across the entire manuscript. I attached a revised PDF.

Please clarify all the points above so the reader can understand clearly what was done and based on what  --- existing --- methodologies.

 Please note the word THE before the monastery name is necessary across the entire manuscript. I attached a revised PDF.

Author Response

Reviewer coment

Answer

The work presented is solid but based on a scarce number of individuals. Which is understandable given the ancestry and rarity of the specimens.

However, the method used is NOT developed by the authors for the first time in this paper as the narrative and abstract seems to indicate. Microsatellite locus T8 was not detected by the authors in this paper, it was previously found and reported in other papers, so they are using a method that exists already. The narrative has to reflect this clearly. They are using an existing method. Not a novel method.

Yes, Tc8 was not detected for the first time in this paper.

However, in our previous study, the aim was not hybridisation between Tilia species, but assessment of genetic structure and diversity. In this study, we used the Tc8 locus with the aim to solely test the hybridization and therefore it is original and different from our earlier study on Tilia.

In MM section we clearly say that we used the Tc8 locus that in a previous study was a putative marker for distinguishing between the Tilia species.

We used Tc8 on the new material to study the hybridisation between Tilia species.

The way it is written seems to indicate you developed it and present it for the first time in this paper  which is incorrect. Danusevicius (12-13) did this before you and called it Tc8. I am not sure why you use T8 here and then refer Tc8 for the same test by Danusevicius (12-13). If it is the same loci you should use the same nomenclature, Tc8.  It doesn't matter he is an author in the current paper. The method already exists. It is not novel in this paper.

As regards T8, it was a misprint, it should be Tc8, we corrected that thought out the manuscript.

We do not say that Tc8 was developed in the present study. We revised the 1st paragraph of the MM section formulate is as follows:

While carrying out a country-wide genetic study on 543 adult T. cordata trees in Lithuania based on genomic microsatellite markers [12], we identified a locus Tc8 unambiguously discriminating between T. cordata, T. platyphyllos. The set of microsatellite loci was developed for genetic studies of T. cordata by [12,22]. Danusevicius et al. [13] found that the locus Tc8 was fixed to a 140 bp allele in T. cordata (541 sampled individuals) and amplified alleles above 160 bp in two trees with T. platyphyllos-like morphology (sampled in a national park). In the present study, our intention was to use this Tc8 locus to investigate the taxonomy of six old- growth Tilia trees growing in the Pazaislis monastery yard...“.

It is difficult to know the number of replicates for each measurement or molecular test as they are not listed across the entire paper. Authors should tell the reader to check Table 3 or list (N=xxx) when speaking about measurements or genetic profiles. We need to know how many times something was done to give statistical support.

The DNA genotyping was done without replicates, because the capillary electrophoresis profiles on the DNA sequencer were clear and reliable for each individual at the Tc8 locus as well as for the six old growth individuals at all 14 nuclear microsatellite locus.

In the Table 3, we used the total number of bracts and leaves of all the investigated trees. The total number of trees and the number of measured bracts and leaves per tree are described in the section 4.3. We have corrected the word “Number” to “Total number of bracts” and “Total number of leaves”.

Part of the problem to understand the paper is that it is very succinct and mostly refers to methodologies of the authors former paper for the genetics. The genetic methods are described in half a line “The DNA extraction, PCR and capillary electrophoresis procedures are described in [12]. Nothing else. Not very clear I would say. At least write: briefly, ….” and describe some of the methods.

Also, what happened to the other 13 microsatellite listed in the abstract? They don’t appear anywhere in the paper. I assume this is what you call “genotyped”? and where are the methods for this? Are the 13 microsatellites across the entire genome? this is important information. And where is this T8 located??? nuclear, mitochondria? I assume nuclear but nothing is mentioned in the text. Depending on this the results are very different if it is mitochondria. That would be heteroplasmy, not an hybrid. This is crucial information.

Ok, we briefly expanded on the DNA analyses. However, we suggest following the journal’s strategy to be concise and do not repeat what is available elsewhere in an open access publication.

Thanks for good remarks. We fully agree. In the 1st paragraph of MM and abstract, we clearly specified that the 14 microsatellite loci (including the TC8) are nuclear microsatellite loci. The 14 loci were used to study genetic associations between the six old growth T. cordata trees from Pazaislis. We explained this in the MM section by noting that the detail genomic characteristics of the microsatellite loci is given in the study 12 and the refences therein)

It is simply too succinct and we don’t know how the molecular part of the work was done. The morphometry is far better described.

See above we expanded on DNA part.

Please note the word THE before the monastery name is necessary across the entire manuscript. I attached a revised PDF.

OK we added that.

Please clarify all the points above so the reader can understand clearly what was done and based on what  --- existing --- methodologies

Ok WE DID SO

Comments on the Quality of English Language Please note the word THE before the monastery name is necessary across the entire manuscript. I attached a revised PDF

OK, English was revised.

Round 2

Reviewer 3 Report

The manuscript has been much improved.

Some minor english revisions:

"revealed the all six old-growth" - revealed that all six....

"Additionally, the genotyping at a set of 14 nuclear" -- Additionally, the genotyping of a set of 14 nuclear.....

bee keeping - beekeeping  (just one world}

but it is most widespread in - but it is widespread in....

We also used morphological descriptions of linden by[35,38]  - needs a space by [35,38].

What is  linden species? the term is used without description. Needs a description. For example - We also used morphological descriptions of species of linden (Tilia L. spp.) by (35, 38) in our assessments....

just like one of your references does : Weryszko-Chmielewska, E.; Sadowska, D.A. The Phenology of flowering and pollen release in four species of linden (Tilia L.). Journal of Apicultural Science 2010, 54, 99–108.

just minor typos

Author Response

Reviewer comment

Answer

Some minor english revisions:

Thank you for your revision. We have revised the manuscript based on all your comments:

"revealed the all six old-growth" - revealed that all six....

22 line

"Additionally, the genotyping at a set of 14 nuclear" -- Additionally, the genotyping of a set of 14 nuclear…

26 line

bee keeping - beekeeping  (just one world}

45 line

but it is most widespread in - but it is widespread in....

49 line

We also used morphological descriptions of linden by[35,38]  - needs a space by [35,38].

320 line

We also used morphological descriptions of linden by[35,38]  - needs a space by [35,38].

What is  linden species? the term is used without description. Needs a description. For example - We also used morphological descriptions of species of linden (Tilia L. spp.) by (35, 38) in our assessments....

just like one of your references does : [18] Weryszko-Chmielewska, E.; Sadowska, D.A. The Phenology of flowering and pollen release in four species of linden (Tilia L.). Journal of Apicultural Science 2010, 54, 99–108.

321 line
